# Polyamine regulation of ion channel assembly and implications for nicotinic acetylcholine receptor pharmacology

Madhurima Dhara[1], Jose A. Matta[1], Min Lei[1], Daniel Knowland[1], Hong Yu[1], Shenyan Gu[1] & David S. Bredt [1⊠]

Small molecule polyamines are abundant in all life forms and participate in diverse aspects of cell growth and differentiation. Spermidine/spermine acetyltransferase (SAT1) is the rate-limiting enzyme in polyamine catabolism and a primary genetic risk factor for suicidality. Here, using genome-wide screening, we find that SAT1 selectively controls nicotinic acetylcholine receptor (nAChR) biogenesis. SAT1 specifically augments assembly of nAChRs containing α7 or α4β2, but not α6 subunits. Polyamines are classically studied as regulators of ion channel gating that engage the nAChR channel pore. In contrast, we find polyamine effects on assembly involve the nAChR cytosolic loop. Neurological studies link brain polyamines with neurodegenerative conditions. Our pharmacological and transgenic animal studies find that reducing polyamines enhances cortical neuron nAChR expression and augments nicotine-mediated neuroprotection. Taken together, we describe a most unexpected role for polyamines in regulating ion channel assembly, which provides a new avenue for nAChR neuropharmacology.

[1] Neuroscience Discovery, Janssen Pharmaceutical Companies of Johnson and Johnson, 3210 Merrifield Row, San Diego, CA 92121, USA.
⊠email: dbredt@its.jnj.com

Neuronal nicotinic acetylcholine receptor (nAChR) ion channels mediate behavioral, cognitive and autonomic effects of acetylcholine and addictive properties of nicotine[1–6]. The nAChR family comprises nine alpha (α2-10) and three beta (β2-4) subunits that combine to form an array of pentameric cation channels throughout the brain and peripheral nervous systems. The major nACh receptors in brain are α7 homomers, α4β2 heteromers, and α6-containing heteromers. These receptors are targets of numerous approved and experimental medicines for diverse conditions including Alzheimer's disease, Parkinson's disease, and neuropathic pain[7,8].

Whereas nAChRs are compelling drug targets, progress is hampered because most receptor subtypes do not functionally express in the non-neuronal cell lines used for screening[9,10]. We previously identified NACHO, a four-pass transmembrane protein that serves as a client-specific chaperone for assembly of most neuronal nAChRs[11–13]. In brain, NACHO is essential for function of α7 receptors, and additional proteins, Ric-3[14] and certain Bcl-2 family members[15], synergize with NACHO to enhance α7 assembly and surface trafficking. NACHO also promotes assembly of α4β2 receptors, though Ric-3 and Bcl-2 proteins have minimal impact on α4β2.

Here, we sought factors that conspire with NACHO to enhance function of α4β2 receptors, the most abundant nAChR subtype in brain. These neuronal α4β2 receptors mediate diverse physiological actions of acetylcholine and underlie nicotine dependence[16]. Genome-wide cDNA screening for protein enhancers of α4β2 function identified, quite surprisingly, a single standout clone that encodes spermidine/spermine acetyltransferase (SAT1), the rate limiting enzyme for polyamine catabolism. By degrading polyamines, SAT1 promotes α4β2 function to an even greater extent than does NACHO.

Polyamines are abundant polycations, including spermidine and spermine, that play multiple roles in cell growth, differentiation and survival[17]. The interplay between their synthesis by ornithine decarboxylase-1 (ODC1) and their degradation by SAT1 controls polyamine levels. ODC1 is amongst the most dynamically regulated of all human proteins, and ODC1 is a drug target in oncology and infectious disease[17,18]. SAT1 transcription is also highly-regulated, and its acetylation of polyamines promotes their cellular export[18]. Interestingly, numerous large genomic studies link polymorphisms in SAT1 with suicidal behavior[19]. In neurons, polyamines play important roles in synaptic transmission by conferring inward rectification to certain potassium channels, AMPA receptors and nACh receptors[20–22]. Polyamines also participate in the pathogenesis of neurodegenerative disorders[17] and the excitotoxicity associated with cerebral ischemia[23].

We now find that polyamines also control assembly of neuronal α4β2 and α7 receptors. By contrast, polyamines do not modulate assembly of α6β4 nAChRs, AMPA receptors or any other ion channel tested. Whereas polyamines classically regulate channel gating by occluding the ion pore, polyamine regulation of nAChR assembly instead relies on negatively charged residues within the α4 or α7 cytosolic loop. Neuropharmacology studies using wild-type and NACHO knockout mice show that lowering polyamine levels selectively upregulates cerebrocortical α4β2 and α7 levels and enhances the neuroprotective properties of nicotine. These studies identify an unexpected role for polyamines in controlling ion channel biogenesis and suggest new strategies in neuropharmacology.

## Results

### Genome-wide screening identifies SAT1 as an enhancer of nAChR.

To identify novel regulators of α4β2 receptors, we co-transfected human embryonic kidney 293T cells (HEK293T) with plasmids encoding α4 and β2 subunits along with individual constructs from a 5943-cDNA clone library from the Broad Institute[24]. Transfected cells were stimulated with nicotine (100 μM) and intracellular $Ca^{2+}$ was quantified with a fluorescence imaging plate reader (FLIPR) (Fig. 1a). As previously shown[11], α4β2 alone produced a small nicotine-evoked $Ca^{2+}$ signal, and co-transfection with NACHO significantly enhanced this (Fig. 1a). High throughput screening identified a single clone that profoundly augmented the nicotine-induced $Ca^{2+}$ response—to levels much higher than NACHO; this clone encoded spermidine/spermine N1-acetyltransferase (SAT1) (Fig. 1a).

We evaluated the effect of SAT1 on other nAChRs and other Cys-loop receptors both by FLIPR and electrophysiology. Consistent with our screening results, SAT1 dramatically increased ACh-evoked currents from α4β2 (Fig. 1c, d). SAT1 also synergized with NACHO to further enhance the α4β2-mediated $Ca^{2+}$ influx (Fig. 1b, d), suggesting that SAT1 and NACHO employ different mechanisms. Whereas SAT1 alone did not rescue homomeric α7 function, SAT1 powerfully synergized with NACHO to increase α7 mediated currents (Fig. 1b–d). In contrast, SAT1 had no significant effect on α6β4 or 5-HT3A receptor function (Fig. 1b–d).

SAT1, a small cytosolic protein, is the rate-limiting enzyme for polyamine catabolism (Supplementary Fig. 1). Together with polyamine oxidase, SAT1 acetylates higher-order polyamines converting them to inactive forms that are transported out of cells[18]. By lowering polyamine levels, SAT1 effects on nAChRs could reflect disinhibition, as polyamines can negatively affect nAChR gating[22]. This seemed unlikely for two reasons, First, α6-containing nAChRs are more sensitive to polyamine inhibition of gating than are α7[25] whereas we find the opposite sensitivity to SAT1 in our functional assays (Fig. 1b–d). Second, polyamines do not block nAChRs at the hyperpolarized membrane potentials we used for patch clamp studies[22]. As an alternative mechanism, we asked whether SAT1 increases nAChR surface expression. To assess this, we utilized extracellular HA-tagged subunits that allow detection of surface receptors without disrupting channel function[11]. Strikingly, SAT1 boosted surface levels of both α4β2 and α7 (Fig. 2a, b). SAT1 also augmented surface trafficking atop the effects of the α7 protein chaperones Ric-3, Bcl-$X_L$, and Mcl-1, as well as the α7 chemical chaperone/orthosteric antagonist, methyllycaconitine (MLA) (Supplementary Fig. 2a, b) implying that receptor upregulation by SAT1 involves a mechanism distinct from any previously described.

### SAT1 promotes assembly of nAChRs by catalyzing polyamines.

To determine whether effects on nAChRs involve the catalytic activity of SAT1, we constructed a Tyr140Phe mutant that abolishes SAT1 enzyme activity[26]. This mutant SAT1 (mutSAT1) did not change surface expression of α4β2 or α7 receptors in the presence or absence of NACHO (Fig. 2a, b). Ornithine decarboxylase 1 (ODC1) is the rate-limiting enzyme for polyamine synthesis (Supplementary Fig. 1), and difluoromethylornithine (DFMO) is an ODC1 active site inhibitor[27] that depletes cellular polyamines[18]. Like SAT1 co-transfection, DFMO pretreatment augmented surface expression of α4β2 and synergized with NACHO to further enhance α4β2 and α7 surface receptors (Fig. 2a, b). Pursuing the opposite tact, we found that co-transfection with the polyamine biosynthetic enzymes ODC1 and adenosylmethionine decarboxylase 1 (AMD1) diminished nAChR surface staining (Supplementary Fig. 3a, b). Neither catabolic (SAT1 co-transfection or DFMO pretreatment) nor anabolic (ODC1 and AMD1 co-transfection) polyamine manipulations affected surface expression of the 5-HT3A receptor (Fig. 2a, b; Supplementary Fig. 3a, b).

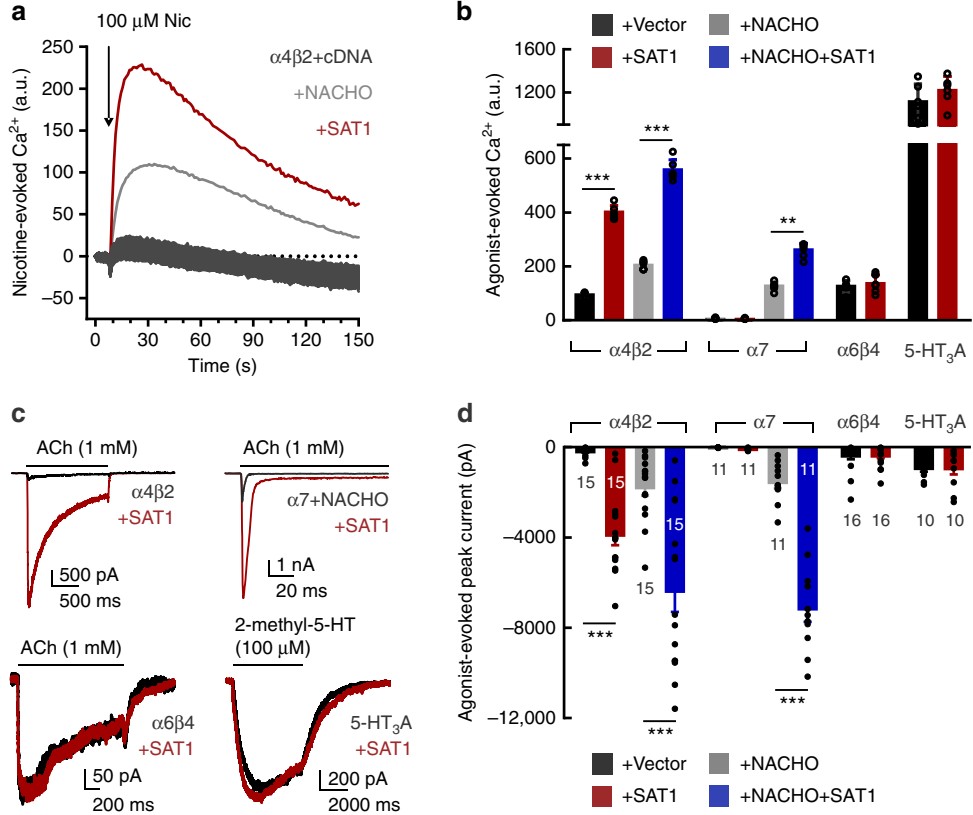

**Fig. 1 High throughput cDNA screening for enhancers of α4β2 function identifies SAT1. a** HEK293T cells in 384-well plates were co-transfected with α4-expressing and β2-expressing plasmids along with an individual cDNA (black traces) from a human ORF collection (Broad Institute). Cells were stimulated with 100 μM nicotine (Nic). The highest response occurred in the well containing SAT1 (red trace) and was twice the α4β2 + NACHO response (gray trace), which served as positive control. **b** Quantification (mean ± SD) of maximum $Ca^{2+}$ signal upon agonist stimulation from HEK293T cells transfected as indicated ($n = 6$). Co-transfection of SAT1 significantly enhances $Ca^{2+}$ signal in α4β2 ($p < 1e^{-4}$), α4β2+NACHO ($p < 1e^{-4}$) and α7+NACHO ($p = 0.002$) cells. For all α6β4 conditions, cells were co-transfected with plasmids encoding accessories BARP and SULT2B1[13]. **c** Representative whole-cell current responses elicited from HEK cells co-transfected with GFP and cDNAs as indicated. **d** Summary graphs of agonist-evoked peak currents (mean ± SEM) from HEK293T cells transfected with indicated cDNA combinations. Similar to maximum $Ca^{2+}$ signal, co-transfection of SAT1 boosts evoked currents mediated by α4β2 ($p < 1e^{-4}$), α4β2+NACHO ($p < 1e^{-4}$), α7+NACHO ($p < 1e^{-4}$) but not α6β4 ($p = 0.98$) or 5-HT₃A ($p = 0.73$). Numbers indicate number of transfected cells that were analyzed and were pooled from three independent cultures. **p < 0.01, ***p < 0.001, One-way ANOVA between the groups for α4β2 and α7. Mann–Whitney $U$ test versus control for α6β4 and 5-HT₃A. Source data for panel **b** and **d** are provided as a Source Data file.

We asked next whether polyamines influence nAChR assembly, which can be probed with orthosteric ligands, such as [³H] epibatidine, that only bind at the interface between folded subunits[28]. Remarkably, either SAT1 or DFMO enhanced [³H] epibatidine binding to α4β2 or α7 receptors (Fig. 2c). Furthermore, ODC1 and AMD1 reduced ligand binding to these receptors (Supplementary Fig. 3c).

N1,N11-bis-(ethyl)-norspermine (BenSpm) is a stable poly-amine analog and SAT1 inhibitor[26,29], and treatment of cells with 10 μM BenSpm specifically reversed the effects of SAT1 on nAChR surface expression (Fig. 2d). By contrast, incubating cells with spermine, which does not penetrate cells and is rapidly degraded by SAT1, did not affect nAChR surface expression (Fig. 2d). Taken together, we find that polyamines, independent of their established role in ion channel gating, enhance nAChR function by promoting surface expression and augmenting receptor assembly.

**Polyamines regulate nAChRs assembly via their cytosolic loop.** Certain neuronal nAChRs display blunted ion flow at depolarized potentials. This inward rectification is mediated by polyamine binding to pore-lining glutamate residues in the nAChR

transmembrane domain 2 (TM2) corresponding to Glu247 in α4[22,30]. It was previously shown that mutating this acidic residue to alanine (α4_E247A) relieves polyamine block of α4β2 channels in *Xenopus laevis* oocytes[22]. Loss of negative charge in the α4_E247A also reduces calcium permeability through the mutant receptor. Accordingly, we found that a E247A α4 mutant (Fig. 3a) co-transfected with β2 evinced minimal nicotine-evoked $Ca^{2+}$ influx in HEK293T cells (Fig. 3b), and this was unaffected by preincubation with DFMO or co-transfection with SAT1. By contrast, DFMO or SAT1 enhanced surface expression of α4_E247A/β2 similar to wildtype α4β2 (Fig. 3c, d). These data establish distinct mechanisms for polyamine regulation of nAChR gating and trafficking.

We next mutated Trp156 (Fig. 3e) in the α4 ligand-binding domain[30]. As expected, when co-transfected with β2, this α4_W156A mutant was functionally inactive and did not bind to [³H]epibatidine (Fig. 3f, g). By contrast, this mutant showed typical surface staining enhancement upon co-transfection with SAT1 cDNA (Fig. 3h, i) indicating that polyamine regulation is independent of agonist binding.

To identify nAChR regions responsible for regulation, we generated chimeras of α4 with α6, as surface expression of the latter is not regulated by polyamines (Fig. 4a). We co-transfected

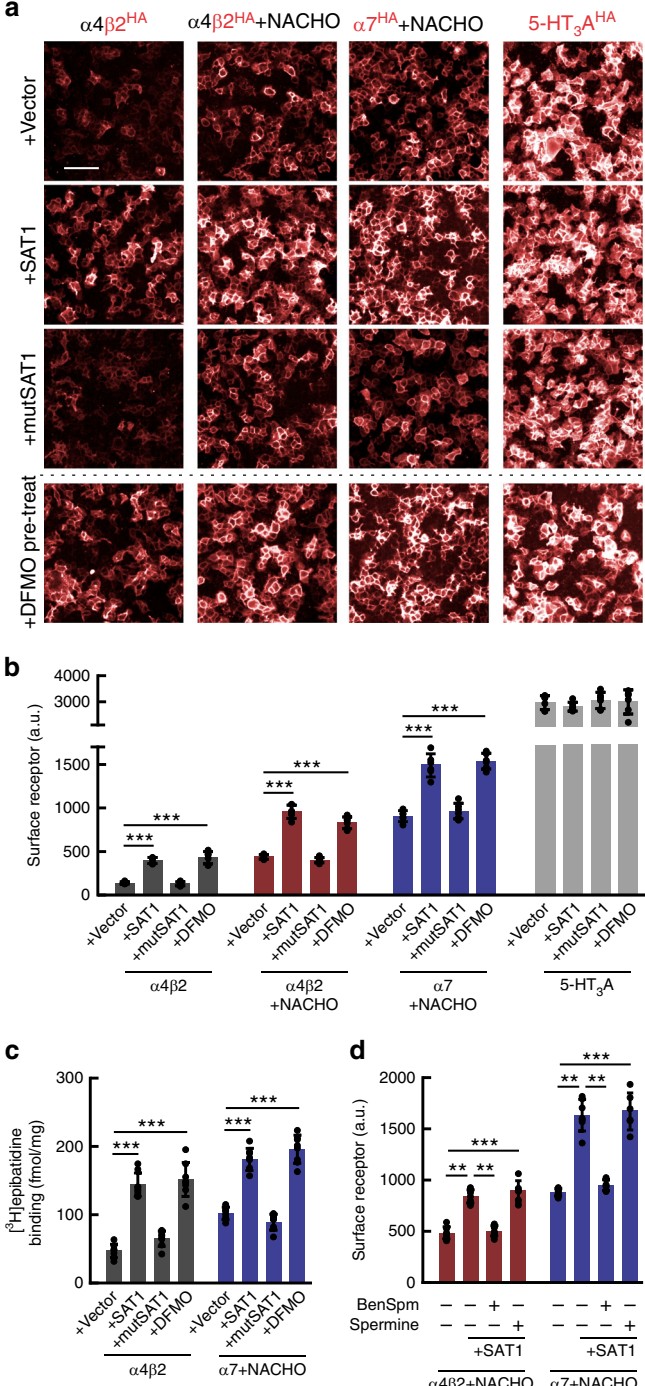

**a**

**b**

**c**

**d**

**Fig. 2 By catalyzing polyamines, SAT1 promotes surface expression and assembly of nAChRs. a** HEK293T cells were co-transfected with cDNAs encoding HA-tagged receptors and other plasmids as indicated. Some cells were pre-treated for 24 h with 1 mM DFMO. Immunofluorescent labeling of the extracellular HA-tag in unpermeabilized cells enabled visualization (red) of surface receptors. Scale bar = 50 μm. **b** Summary graph quantifies surface HA-labeling ($n = 6$). Co-transfection with wildtype SAT1, not its mutant version, promotes surface expression of α4β2 ($p < 1e^{-4}$), α4β2+NACHO ($p < 1e^{-4}$) and α7+NACHO ($p < 1e^{-4}$) but not 5-HT$_3$A ($p = 0.84$). Pre-treating cells with DFMO show comparable results. **c** Quantification of [$^3$H]epibatidine binding to HEK293T cell membranes transfected and pre-treated with DFMO where indicated ($n = 8$). Similar to surface HA-labeling, SAT1 co-transfection and DFMO preincubation increase radioligand binding to α4β2 ($p < 1e^{-4}$) and to α7+NACHO ($p < 1e^{-4}$). **d** Quantification of surface α4β2-HA or α7-HA in HEK293T cells transfected with SAT1 and pretreated with BenSpm ($n = 7$) or spermine ($n = 6$) as indicted. Only BenSpm occludes SAT1 mediated increase of surface α4β2 ($p = 1e^{-3}$) or α7 ($p = 1e^{-3}$). Data in **b-d** are presented as mean ± SD. **p < 0.01, ***p < 0.001, One-way ANOVA between the groups. Source data for panel **b-d** are provided as a Source Data file.

sites on α7 (Supplementary Fig. 4e, f), which demonstrates that the intracellular loop determines polyamine-regulation of both α4 and α7 nAChR assembly.

Previous studies noted that, within the T3-TM4 loop, a subdomain proximal to TM4 contains conserved positively and negatively charged residues[31]. Interestingly, we observed certain negatively charged residues (Glu and Asp) in this subdomain that are conserved between α7 and α4 but diverge in α6 to uncharged Asn (Supplementary Fig. 5a, b)[31]. Mutating those Glu and Asp to Ala in α7 (E$_{438}$A or D$_{451}$A) and α4 (E$_{569}$A or D$_{582}$A) blocked SAT1-mediated upregulation of α7 assembly (Supplementary Fig. 5c, d) and α4β2 surface expression (Supplementary Fig. 5 e, f). By contrast, these mutations did not blunt effects of Ric-3 on α7 or NACHO on α4β2. Replacing the corresponding Asn residues (N$_{434}$ or N$_{447}$) to Glu within the α6 cytosolic loop conferred moderate but significant SAT1 and DFMO sensitivity to α6β4 receptors (Supplementary Fig. 5g, h). These experiments identify negatively charged residues within the TM3-TM4 cytosolic loop that can mediate polyamine regulation of nAChR assembly and surface expression. Future studies are needed to fully elucidate these mechanisms.

**Polyamines regulate nAChR gating and assembly independently.** To further distinguish mechanisms for polyamine control of nAChR assembly and gating, we compared effects of the membrane-permeant polyamine analog BenSpm with philanthotoxin-343 (PhTx-343), which cannot enter cells and engages the channel pore via a long polyamine tail[25,32]. Neither BenSpm nor PhTx-343 preincubation changed basal surface levels of α4β2. Whereas BenSpm abrogated SAT1-mediated enhancement of surface α4β2, PhTx-343 did not (Fig. 5a, b). By contrast, both BenSpm and PhTx-343 preincubation abolished nicotine-evoked Ca$^{2+}$ with or without SAT1 co-transfection (Fig. 5c). These data demonstrate that only cell-permeant polyamines can control of nAChR surface trafficking.

To more completely elucidate these mechanisms, we compared effects of BenSpm and PhTx-343 on α4-containing and α6-containing receptors, as SAT1 regulates trafficking of α4β4 but not α6β4 (Fig. 4b). As expected, pre-treatment with BenSpm, but not PhTx-343, reversed SAT1-mediated enhancement of α4β4 surface expression and neither had effect on α6β4 surface expression (Fig. 5d, e). By contrast, both BenSpm and PhTx-343

these α4/α6 chimeras with β4 and quantified both receptor surface expression and nicotine-evoked Ca$^{2+}$ signaling (Fig. 4). SAT1 co-transfection enhanced a chimera containing the α6 N-terminal and α4 C-terminal region but did not affect the converse chimera (Fig. 4a–c), which implies that polyamine regulation involves the channel or the large cytosolic loop, which occurs between TM3 and TM4. We next constructed a pair of chimeras that swapped only this cytosolic loop. Quantification of receptor surface expression and function showed that the α4 TM3-TM4 cytosolic loop is both necessary and sufficient for SAT1-dependent regulation of these nAChRs (Supplementary Fig. 4). Similarly, exchanging the TM3-TM4 loop of homomeric α7 receptor with that of α6 (Supplementary Fig. 4d) abolished SAT1-mediated upregulation of alpha-bungarotoxin (α-Bgt) binding

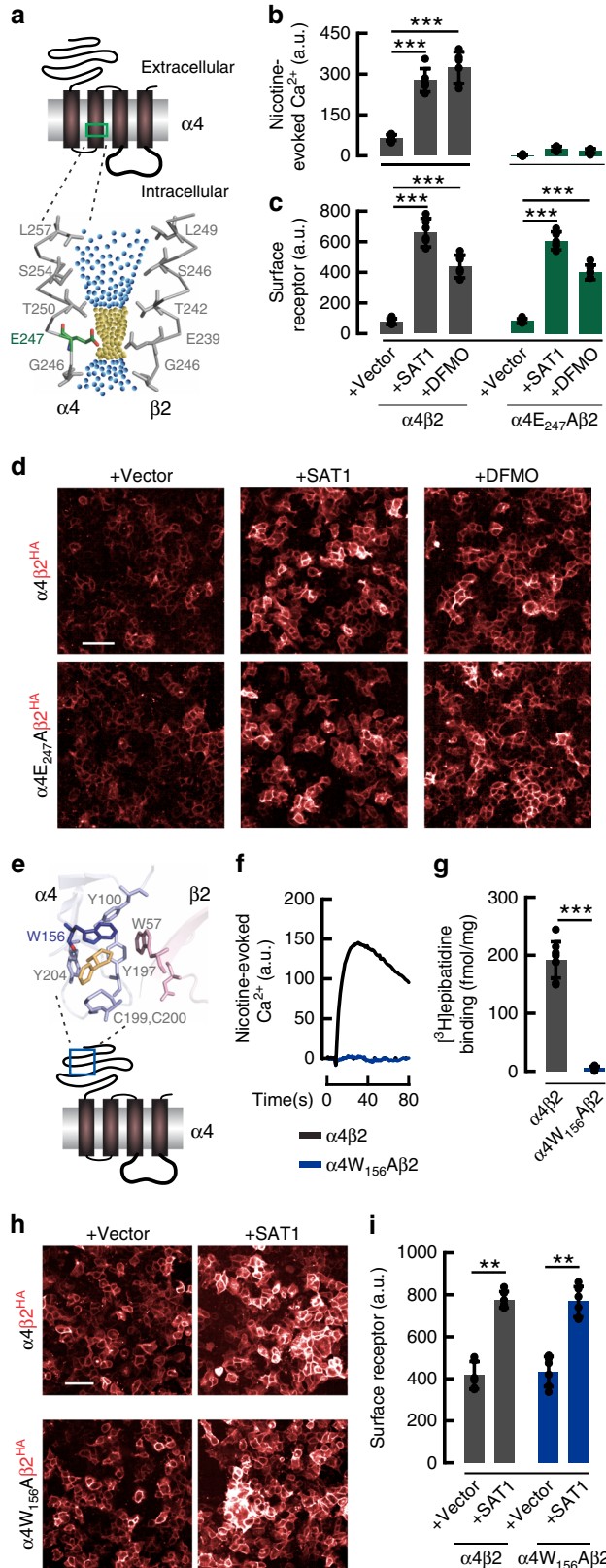

**Fig. 3 Polyamine regulation of α4β2 assembly is mechanistically distinct from channel gating or agonist binding. a** Cartoon depicting α4 nAChR subunit. Enlarged view (bottom) highlights Glu247 in α4 TM2 critical for polyamine regulation of gating and $Ca^{2+}$-permeability. Yellow spheres represent pore diameter >2.8 Å, blue spheres >5.6 Å. (PDB: 5KXI) **b** Quantification shows wild-type α4β2-mediated peak nicotine-evoked $Ca^{2+}$ is enhanced in cells co-transfected with SAT1 ($p < 1e^{-4}$) and pre-treated with DFMO ($p < 1e^{-4}$). $α4E_{247}Aβ2$ transfected cells have reduced nicotine-evoked $Ca^{2+}$ ($p = 0.01$), which is not changed with SAT1 or DFMO ($n = 6$). **c** Summary graph ($n = 6$) and **d.** representative images of fluorescent anti-HA labeling of non-permeabilized HEK293T cells transfected as indicated and pretreated with DFMO where noted (Scale bar = 50 μm). Quantifications are displayed as mean ± SD. ***$p < 0.001$, One-way ANOVA between the groups. **e** Cartoon depicting α4 nAChR subunit. Enlarged view (top) highlights Trp156 in α4 that forms a cation-π interaction with nicotine (yellow). **f** Nicotine-evoked $Ca^{2+}$ from wild-type α4β2 (black trace) or mutant $α4W_{156}Aβ2$ (blue trace) (PDB: 5KXI). **g** Binding of [³H]epibatidine (10 nM) to HEK293T cell membranes is significantly reduced ($p < 1e^{-5}$) in mutant $α4W_{156}Aβ2$ co-transfected group compared with wild-type α4β2 ($n = 8$). **h** Fluorescent anti-HA labeling of non-permeabilized HEK293T cells transfected as indicated. Scale bar = 50 μm. **i** Quantification of surface labeling from **h** ($n = 6$). SAT1 enhances surface labeling of both α4β2 ($p = 0.002$) and $α4W_{156}Aβ2$ ($p = 0.002$) receptors. NACHO cDNA was included for all receptor transfections in **f–i**. Quantifications are means ± SD. **$p < 0.01$, ***$p < 0.001$, Mann–Whitney U test versus control was used for data shown in panel **g**. Source data for panels **b**, **c**, **g** and **i** are provided as a Source Data file.

reverse SAT1-mediated enhancement of receptor surface expression, and PhTx-343 did not alter surface expression for any chimera (Fig. 5e and Supplementary Fig. 6d, e). By contrast, acute application of either BenSpm or PhTx-343 blocked nicotine-evoked $Ca^{2+}$-signaling from all receptor chimeras in a concentration-dependent manner (Fig. 5f, Supplementary Fig. 6a–c). These molecular biological and pharmacological studies establish distinct mechanisms and protein domains for polyamine control of nAChR trafficking and gating.

**Polyamines regulate assembly of neuronal nAChRs.** We next explored whether polyamines upregulate endogenous nAChRs in neurons. Accordingly, we treated cultured rat cortical neurons with DFMO and quantified α7 nAChR surface expression with fluorescent α-Bgt. As a positive control, we incubated neurons with 100 μM nicotine[33] and found a significant increase in surface α-Bgt binding sites (Fig. 6a, b). Similarly, treating neurons with DFMO robustly increased surface α-Bgt binding (Fig. 6a, b). By contrast, neither DFMO nor nicotine altered surface expression of the AMPA-type glutamate receptor subunit GluA1 (Fig. 6a, b). Preincubating neurons with DFMO also increased the nicotine-evoked $Ca^{2+}$ signal (Fig. 6c), an effect that was comparable to the enhancement seen following nicotine pretreatment. These nicotine-evoked $Ca^{2+}$ responses reflect α7 activity, as they require the α7-specific positive allosteric modulator (PAM) PNU-12059615[34] and were abolished by the α7-specific inhibitor α-Bgt (Fig. 6c).

[³H]Epibatidine binding in brain largely reflects α4β2 receptors and was absent in cerebral cortex of α4 knockout mice[35]. Importantly, incubating cortical neurons with DFMO or transducing them with a lentivirus expressing SAT1 enhanced levels of [³H]epibatidine binding (Fig. 6d), indicating an increased α4β2 receptor assembly. To confirm polyamine regulation of neuronal α4β2, we transduced cortical neurons with α4-expressing and β2-expressing lentiviruses. Levels of [³H]

application abolished nicotine-evoked $Ca^{2+}$ signaling from either α4β4 or α6β4 (Fig. 5f).

We next evaluated effects of BenSpm and PhTx-343 on the α6/α4 chimeric constructs, which swap the extracellular N-terminal or TM3-TM4 cytosolic loop domains. As predicted, the α4 cytosolic loop was necessary and sufficient for BenSpm pre-treatment to

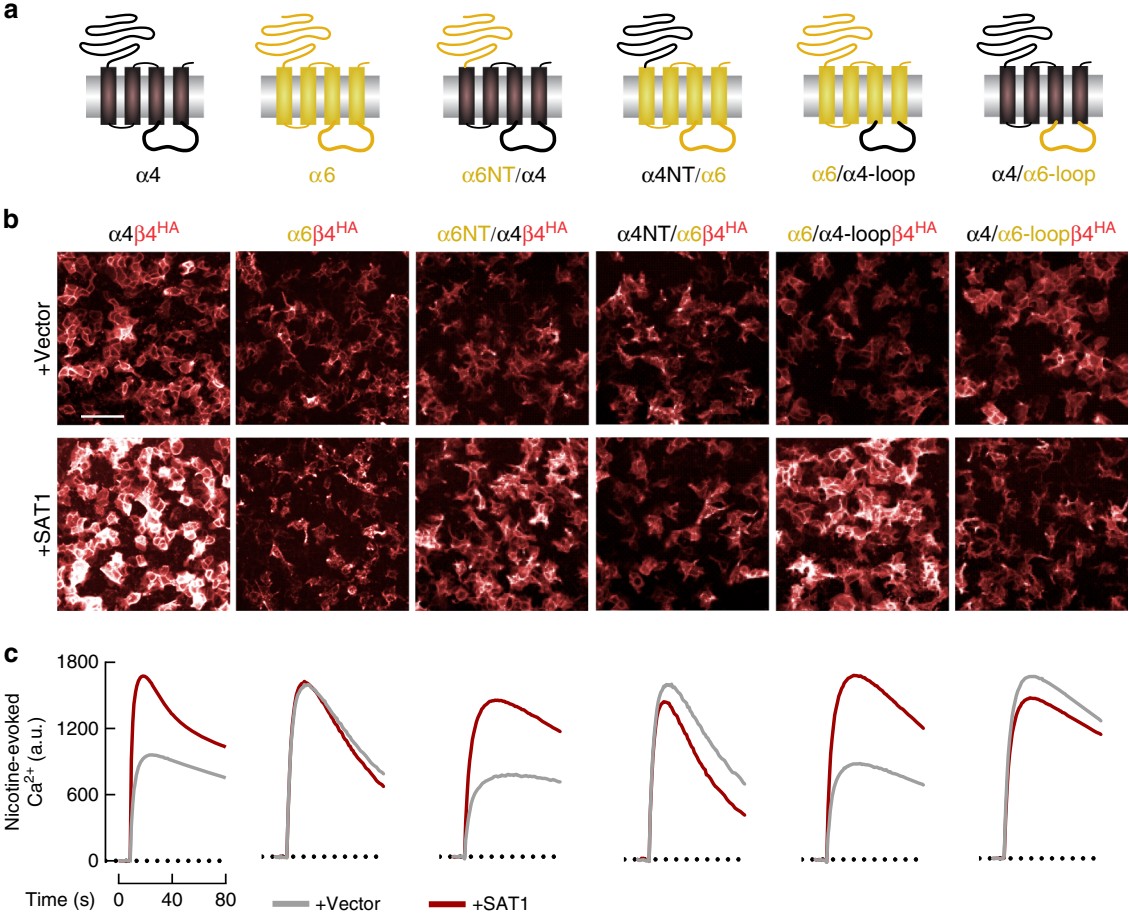

**Fig. 4 The nAChR TM3-TM4 cytosolic loop determines regulation by SAT1. a** Schematics of chimeric α4-α6 constructs. **b** Fluorescent anti-HA labeling of non-permeabilized HEK293T cells co-transfected with cDNAs encoding α6, α4, α4-α6 chimeric constructs and extracellular HA-tagged β4 with or without SAT1 as indicated. All cDNA combinations contained BARP and SULT2B1. Scale bar = 50 μm **c**. Agonist-evoked Ca²⁺ traces from HEK293T transfected as indicated. The α4 cytosolic loop domain is necessary and sufficient for SAT1-mediated regulation for all chimeras. Note that data in **b** and **c** are quantified in Supplementary Figs. 4b, c.

epibatidine binding to the transduced neuronal membranes were ~20 times higher than untransduced neurons, and this was further increased with DFMO pretreatment (Fig. 6d). Importantly, neither DFMO preincubation nor SAT1 transduction altered binding sites for the GABA receptor ligand [³H] flunitrazepam (Fig. 6d). Taken together, these data show that polyamines regulate assembly of neuronal nAChR but not AMPA or GABA receptors.

**Polyamines modulate nicotine-mediated neuroprotection.** Nicotine has well-established neuroprotective properties and can mitigate excitotoxicity in vivo[36] and in vitro[37,38]. This neuroprotection is mediated both by α7[39] and α4β2 nAChRs[38,40]. To assess the functional consequence of polyamine upregulation of receptor assembly, we asked whether this nicotine-mediated neuroprotection could be enhanced by reducing cellular polyamine levels. We induced excitotoxicity by challenging cultured neurons for one hour with glutamate (30 μM) and quantified cell survival and cytosolic cytochrome C (CytC), which is a trigger for apoptosis[41,42]. We also surface-labeled α7 receptors with fluorescent α-Bgt. In line with previous reports, glutamate challenge induced excitotoxicity and increased cytosolic CytC (Supplementary Fig. 7a–c). Co-application of 100 μM nicotine reduced excitotoxicity, but glutamate did not alter surface expression of α7 (Supplementary Fig. 7b, d). On the other hand, neurons pretreated with either DFMO or nicotine showed a three-fold

increase in surface α-Bgt-binding sites (Supplementary Fig. 7a, d) and a concordant reduction of glutamate-induced CytC release (Fig. 6e; Supplementary Fig. 7c). Nicotine or DFMO preincubation also augmented nicotine-mediated protection from glutamate-mediated excitotoxic cell death (Fig. 6e and Supplementary Fig. 7b).

To more directly link the neuroprotective effects of DFMO with nAChRs, we studied NACHO KO mice, which have dramatically reduced nAChR function in brain[12]. We found that nicotine blunted glutamate-induced cell death in neurons from wild-type but not NACHO KO mice (Fig. 7a–c). Fitting with results from our rat cortical neuron experiments, we found that pre-treating wild-type mouse neurons with DFMO increased surface α-Bgt staining (Fig. 7e) and mitigated glutamate-induced increases in cytosolic CytC (Fig. 7d). By contrast, DFMO pretreatment did not augment α-Bgt binding levels in NACHO KO neurons. Accordingly, DFMO preincubation did not confer subsequent nicotine-induced protection from excitotoxic neuronal death in NACHO KO neurons (Fig. 7c). Collectively, these experiments show that polyamines regulate functional assembly of nAChRs, which promotes neuroprotection in wild-type, but not NACHO KO neurons.

**Discussion**
This study identifies an unexpected role for polyamines in controlling assembly of neuronal α4β2 and α7 receptors. Our

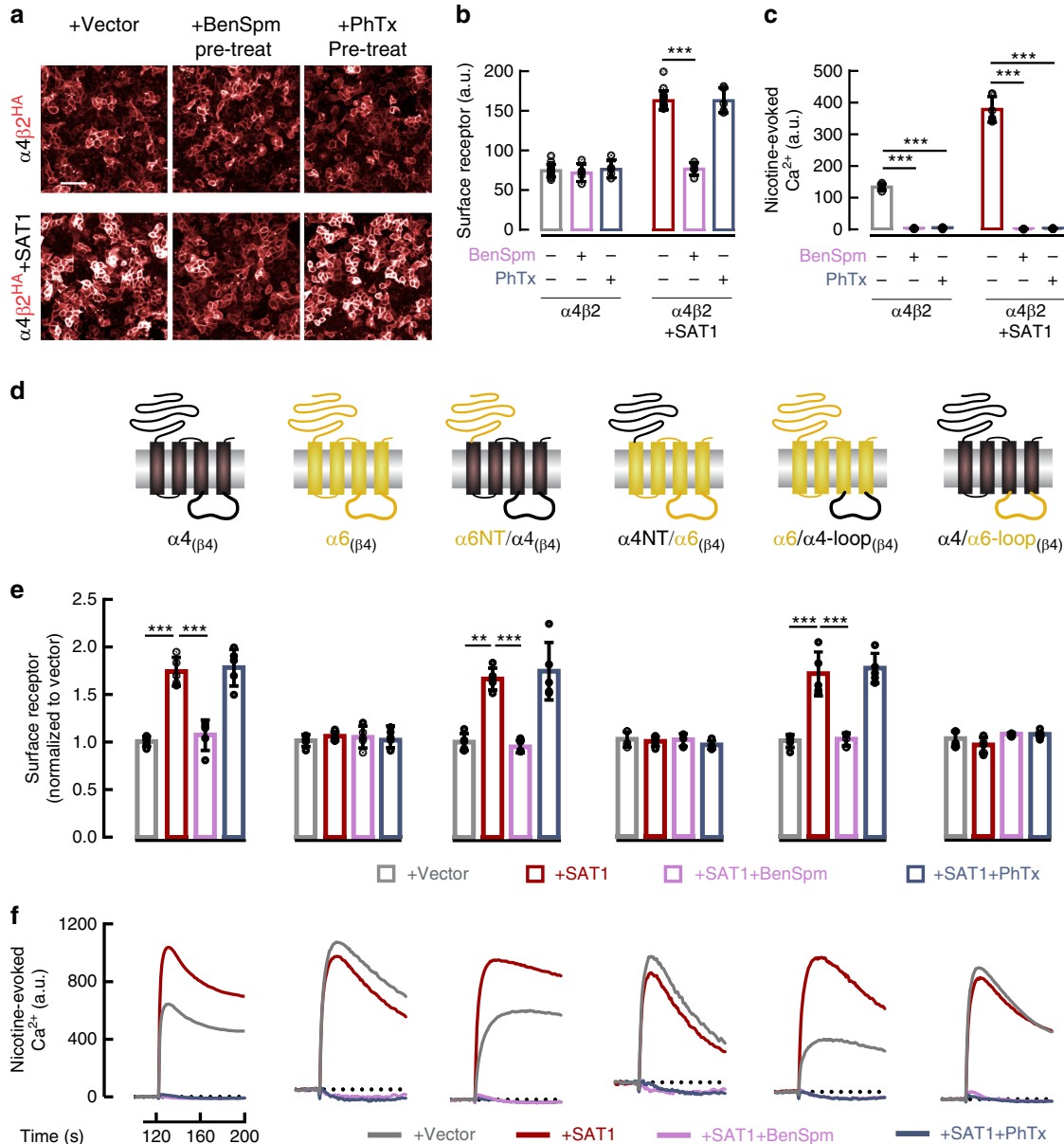

**Fig. 5 Polyamine analogs regulate nAChR gating and assembly by distinct mechanisms. a** Fluorescent anti-HA labeling of non-permeabilized HEK293T cells transfected with cDNAs encoding extracellular HA-tagged β2 and α4 with or without SAT1. Scale bar = 50 μm**. b** Quantification of surface staining for α4β2 receptors with or without SAT1 and BenSpm or PhTx343 pre-treatment ($n = 6$). BenSpm occluded the SAT1-enhanced receptor surface expression ($p < 1\mathrm{e}^{-4}$) but PhTx343 had no effect ($p = 0.86$). **c** Nicotine-evoked $Ca^{2+}$ signal was fully blocked by both BenSpm or PhTx343 ($p < 1\mathrm{e}^{-4}$) ($n = 6$). **d** Schematics of chimeric constructs of α4-α6. **e** Normalized anti-HA surface expression from cells transfected and treated with BenSpm or PhTx343 as indicated ($n = 5$). Only BenSpm occluded SAT1-mediated enhancement of surface receptor expression of α4β4 ($p < 1\mathrm{e}^{-4}$), α6NT/α4β4 ($p < 1\mathrm{e}^{-4}$) and α6/α4-loopβ4 ($p < 1\mathrm{e}^{-4}$). Both SAT1 and BenSpm effects on receptor surface expression required the α4 cytosolic loop. **f** Both BenSpm and PhTx343 fully inhibited nicotine-evoked $Ca^{2+}$ in all receptor combinations. All cDNA combinations included BARP and SULT2B1. Quantifications are displayed as mean ± SD. ***$p < 0.001$, One-way ANOVA between the groups was used for panels **b**, **c** and **e**. Comprehensive dose-response studies are in Supplementary Fig. 5. Source data for panel **b**, **c** and **e** are provided as a Source Data file.

genome-wide analysis found that SAT1 increases assembly of α4β2 receptors to a much greater extent than any other protein screened, including the nicotinic receptor chaperone NACHO. Polyamine-regulated assembly is specific for certain nAChRs, is distinct from other nAChR chaperone mechanisms, and requires negatively charged amino acids within the α4 or α7 cytosolic loop. Blocking polyamine synthesis with DFMO upregulates neuronal α4β2 and α7 surface levels and promotes nicotine-mediated neuroprotection, which provides a new angle for drug discovery.

The biogenesis of pentameric nAChRs is a complex and tightly-regulated process[43]. Seminal studies in the 1980s found that nicotine and other orthosteric ligands upregulate brain nACh protein levels, and this likely participates in nicotine dependence[44,45]. In *C. elegans*, Ric-3 is required for efficient assembly of worm nAChRs[46]; in mammalian brain, NACHO is required for α7[11] and many other nAChRs[12]. Additional proteins synergize with NACHO for assembly of specific mammalian nAChRs. Ric-3[14] and certain Bcl-2 family proteins[15] work with NACHO to promote function of α7, whereas BARP, LAMP5, and

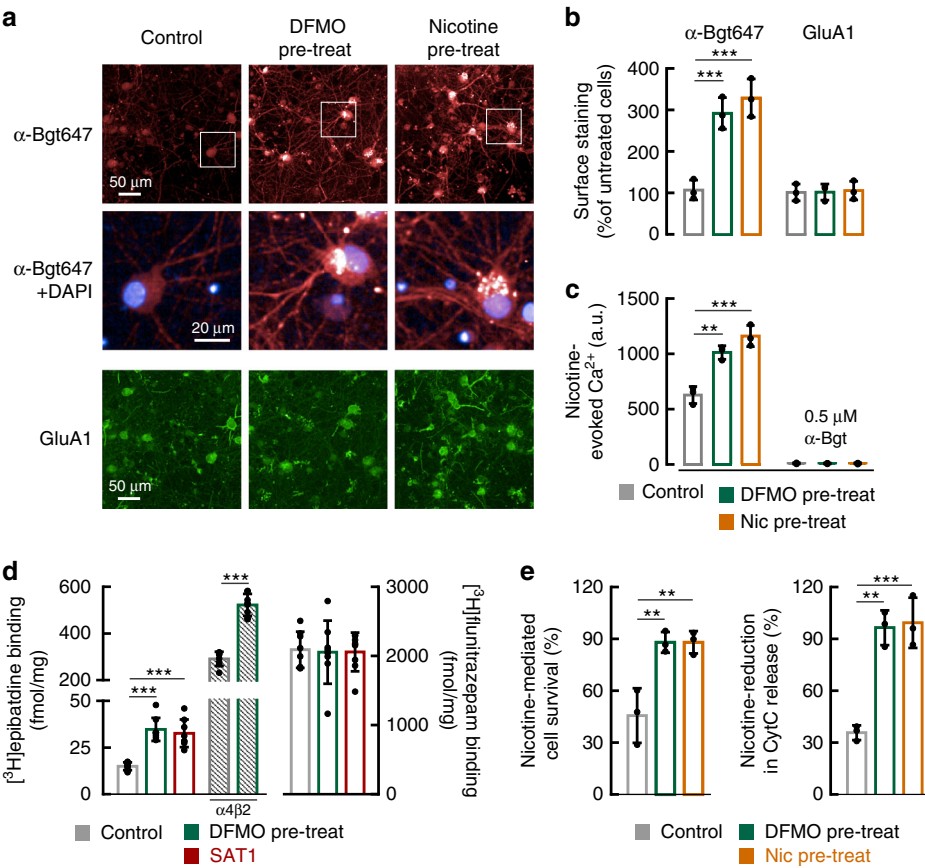

**Fig. 6 DFMO enhances both neuronal nAChRs and nicotine-mediated neuroprotection. a** Fluorescent labeling of non-permeabilized rat cortical neurons (DIV 20) shows upregulation of endogenous α-Bgt674 labeling in neurons pre-treated for six days with DFMO (5 mM) or nicotine (Nic; 100 μM) as indicated. Surface-labeling for α7 used α-Bgt647 (red) and insets (white squares) depict magnifications that include DAPI (blue). Double-labeling with α-GluA1 (green) served as control. **b** Quantification of surface α-Bgt647 shows increased staining with both DFMO and Nic ($p < 1e^{-4}$), but no change was observed in surface GluA1 staining ($p = 0.9$). **c** Summary graph showing nicotine (100 μM) + PNU-12059615- (10 μM) evoked $Ca^{2+}$ from rat cortical neurons (DIV 13) increases when pre-treated for six days with DFMO ($p = 0.002$) or nicotine ($p = 0.0004$). Responses were blocked by α-Bgt (0.5 μM). Data shown in B and C were averaged from 3 independent experiments each containing 6 independent samples. **d** Quantification of [³H]epibatidine (10 nM) and [³H]flunitrazepam (50 nM) binding to membranes from neurons pre-treated with DFMO or transduced with SAT1 show selective increase in [³H]epibatidine binding ($p < 1e^{-4}$). One-way ANOVA compared to control. Neurons transduced with lentivirus expressing α4 and β2 showed much higher levels of [³H]epibatidine binding (hatched bars) ($p < 1e^{-4}$, compared to un-transduced control), and these were further enhanced ($p < 1e^{-4}$) with DFMO treatment ($n = 8$). **e** Acute application of nicotine (gray bars) partially reduced glutamate-mediated cell death (left panel) and mobilization of CytC (right panel). Pre-treating neurons for six days with DFMO (green, 5 mM) enhanced nicotine-mediated cell survival ($p = 0.006$) and decreased CytC mobilization ($p = 0.001$). Nicotine (orange, 100 μM) pre-treatment similarly enhanced nicotine-mediated neuroprotection. Data averaged from 3 independent experiments each containing 6 independent samples. Quantifications are displayed as mean ± SD. **$p < 0.01$, ***$p < 0.001$, One-way ANOVA between the groups for panels **b**–**e**. Source data for panels **b**–**e** are provided as a Source Data file.

SULT2B1 conspire with NACHO to enhance function of α6-containing receptors[13]. Our discovery of SAT1 regulation identifies the polyamine pathway as an additional controlling mechanism for α4β2 and α7 receptors.

These multiple layers of nAChR regulation utilize distinct mechanisms. In the case of α7, NACHO mediates subunit oligomerization[11]. Subsequent regulation by nicotine, Ric-3, and Bcl-2 proteins synergizes with NACHO to promote protein folding, receptor surface expression, and channel function[14,15]. Nicotine and other orthosteric ligands promote assembly through α7's ACh binding domain[45] whereas Bcl-2 proteins[15] and polyamines engage the receptor cytosolic loop. However, polyamine regulation involves a distinct mechanism, as we find SAT1 augments receptor assembly atop NACHO, nicotinic ligand, Ric-3, or Bcl-2 family proteins.

It is intriguing to ask why nAChRs—but not other receptors in the Cys-loop superfamily including 5-HT3 and GABA$_A$ receptors —require accessories for functional expression. One possibility is that multiple assembly mechanisms provide regulatory nodes for controlling nAChR function. NACHO transcription is upregulated by physiological stimuli[47], and Bcl-2 family protein levels are dynamically-induced during developmental and pathological processes[48]. Furthermore, protein accessories may also determine nAChR cellular localization. The α6-containing nAChRs specifically concentrate at presynaptic terminals of specific monoaminergic neurons, and this may be enabled by the lysosomal protein, LAMP5, which displays a similar restricted distribution[49].

Polyamines are ubiquitous in biology and play multiple roles in cell growth, survival, and differentiation[17]. In neurons, polyamines control gating of several important ion channels. Cytosolic polyamines confer inward rectification to certain potassium channels, AMPA receptors, and nAChRs[20–22]. Elegant biophysical studies showed that neuronal depolarization draws cytosolic polyamines into the channel pore, which precludes ion flow[20]. This polyamine site is accessible to extracellular

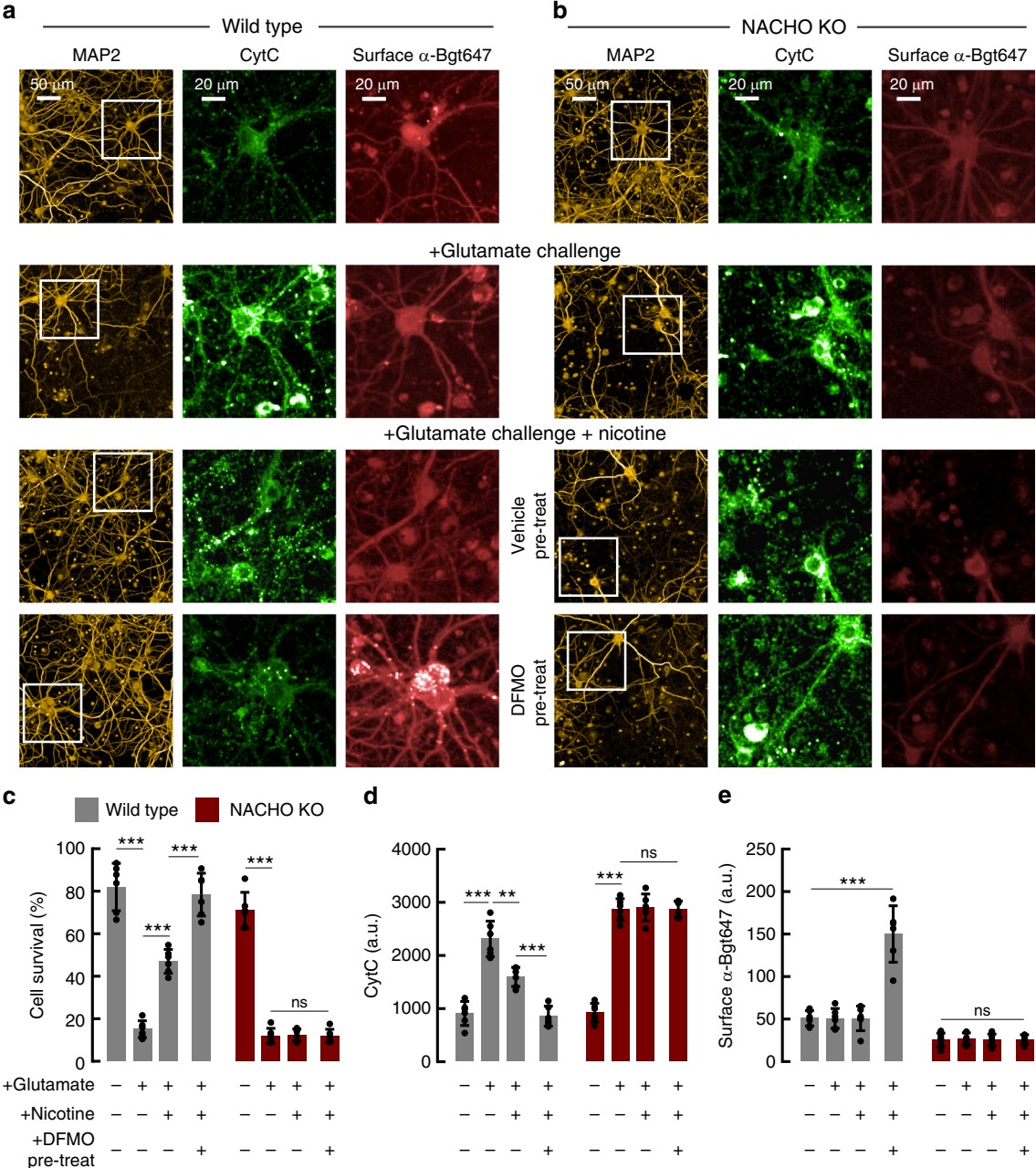

**Fig. 7 DFMO promotes nicotine-mediated neuroprotection in a NACHO-dependent fashion. a, b** Images of cortical neurons (DIV 20) from wild-type (**a**) or NACHO KO (**b**) mice. As indicated, neurons were pretreated with DFMO (5 mM, 4th row) or and challenged with 30 μM glutamate (Glu, 2nd–4th row) in the absence (2nd row) or presence (3rd–4th row) of 100 μM nicotine. The cells were stained for MAP2 (left panel) and cytochrome-C (CytC middle panel) and surface α-Bgt647 (right panel). White squares indicate regions that are magnified. **c–e** Graphs quantify neuronal survival (**c**), CytC (**d**) and surface α-Bgt647 (**e**) ($n = 6$). Acute nicotine increases cell survival ($p < 1e^{-4}$) and reduces CytC mobilization ($p = 0.0002$) during glutamate toxicity in wild-type neurons, but not in NACHO KO neurons ($p = 0.9$). DFMO pre-treatment further promotes nicotine-mediated cell survival ($p < 1e^{-4}$) and reduces CytC mobilization ($p = 0.0002$) in wildtype neurons. DFMO significantly enhances surface α-Bgt647 labeling of wild-type ($p < 1e^{-4}$) but not of NACHO KO neurons ($p = 0.9$). **p < 0.01, ***p < 0.001, n.s = not significant, one-way ANOVA between the groups. Data displayed as mean ± SD. Source data for panel **c–e** are provided as a Source Data file.

philanthotoxins[50], which have long polyamine tails that can engage the channel.

Our studies decisively establish that regulation of α7 and α4β2 assembly by polyamines is distinct from their classical role in controlling ion channel gating. First, we find that SAT1 increases α4β2 and α7 function even when recorded at hyperpolarized potentials that preclude nAChR gating control by cellular polyamines[22] (Fig. 1c, d). Second, SAT1 promotes assembly of nonfunctional α4β2 receptors that cannot bind ACh (Fig. 3e–i).

Third, extracellular philanthotoxin acutely blocks α4β2 receptor function but philanthotoxin does not affect receptor assembly (Fig. 5a–c). Fourth, chronic but not acute application of a cell-permeable polyamine analog, BenSpm, reverses the effects of SAT1 on α7 and α4β2 receptor assembly and function (Fig. 2d). Interestingly, we find that SAT1 does not augment function of α6-containing nAChRs. Taking advantage of this α-subunit specificity, our α6/α4 chimeras determine that polyamine regulation of channel assembly involves the α4 cytosolic loop. Again, this is

distinct from the pore region that determines polyamine control of rectification.

As α7 and α4β2 are the most abundant nAChRs in human brain and control diverse aspects synaptic signaling and plasticity, polyamine regulation of their assembly has important physiological and pathophysiological implications. Polyamine levels in neurons are dynamically regulated over both short-time and long-time scales. Acutely, synaptic transmission increases synthesis of polyamines, which modulate integrative neuronal properties by reducing AMPA receptor currents[51] and provide an excitability buffer by negatively regulating $Na^+$ channels[52]. Long-lasting changes in synaptic transmission associated with pathological processes such as epilepsy dramatically upregulate ODC1[53], which our study predicts would downregulate nAChR assembly. Indeed, α4-containing nAChRs levels are decreased in piriform cortex of kindled mice[54].

Polyamines and nAChRs share compelling links to neuropsychiatric disorders. Numerous genetic, genomic, and biochemical studies identify alterations in the polyamine pathway in major depressive disorder and suicide[55]. An SNP in the promoter region of SAT1 that reduces expression is associated with suicide in a French-Canadian founder population[55]. Postmortem studies find reduced SAT1 protein and mRNA levels in precentral gyrus and cortical frontal lobe[56]. Also, polyamine levels increase during anxiety episodes, and this polyamine stress response pathway[19] could suppress nAChR assembly. Fitting with this, preclinical and early clinical studies suggest that α7 or α4β2 agonists can improve depressive behavior in animal models and in depressed patients[57].

Numerous observations also link polyamine and nAChR alterations with cognitive and neurodegenerative processes. Spermine synthase mutations cause Snyder-Robinson syndrome characterized by mental retardation and juvenile myoclonic epilepsy[58], and these central symptoms are also found in certain patients with mutations in α4 nAChR[59]. Interestingly, both SAT1 overexpression[60] and nicotine[36] protect animals from kainate-induced toxicity. Polyamine levels are elevated[17] and nAChR levels are reduced[61] in Alzheimer's disease, which is best treated today with cholinesterase inhibitors that augment nAChR activity. Furthermore, DFMO[23] and nicotinic agonists[40] both mitigate neurotoxicity from neuronal ischemia. Our results that blocking ODC1 increases neuronal α4β2 and α7 and promotes nicotine-mediated neuroprotection provide a mechanistic link for these observations. As such, targeting this polyamine/nAChR assembly pathway provides a new neuropharmacological strategy.

## Methods

**Genes and molecular biology and cell culture**. The following genes are studied here: (Human forms) CHRNA4 (NM_000744.6), CHRNB2 (NM_000748.2), CHRNA7 (NM_000746.5), CHRNA6 (NM_004198), CHRNB4 (NM_000750), TMEM35A or NACHO (NM_021637.2), Ric-3 (NM_024557.5), Bcl-X$_L$ (NM_138578.2), Mcl-1 (NM_021960.4), SAT1 (NM_002970.3), ODC1 (NM_002539); (mouse forms) 5HT-$_3$A (NM_013561), AMD1 (NM_009665).

Chimeric constructs used in this study, with residues in the parenthesis are: α6NT/α4 (α6$_{1–239}$/α4$_{243–628}$), α4NT/α6 (α4$_{1–242}$/α6$_{240–495}$), α6/α4loop (α6$_{328–465}$ exchanged to α4$_{331–600}$), α4/α6loop (α4$_{331–600}$ exchanged to α6$_{328–465}$), α7/α6loop (α7$_{328–457}$ exchanged to α6$_{339–453}$). In brief, for α6NT/α4 chimera, α4$_{243–628}$ region and α6$_{1–239}$ was linearized and amplified with PCR. The primers for α6$_{1–239}$ contained complementary overhangs immediately upstream and downstream of amplified region. PCR products were ligated using In-Fusion HD Cloning Plus kit (Takara, 683910) in accordance with manufacturer's protocol. All other chimeras were generated following similar strategy and were confirmed by sequencing. Site-directed mutagenesis was done with two complementary primer reactions (primer details provided in Supplementary Table 1), and all mutations were confirmed by sequencing. The α7-HA, β2-HA, and β4-HA constructs contained a PSGA linker and HA tag immediately following the C-terminal residue.

HEK293T cells (ATCC® CRL-3216™) were cultured in DMEM medium supplemented with 10% FBS and 1 mM sodium pyruvate. Cells were seeded at 80-90% confluence and were transiently transfected using FuGENE®6 transfection reagents (Promega Corporation). All functional and expression assays in

HEK293T cells were performed after 48–72 h incubation at 37 °C following transfection, unless stated otherwise.

All animal experiments reported here were overseen and approved by an AAALAC accredited institutional review board. Primary cortical neurons were prepared from E18 rat cortex (supplied by BrainBits®) and E18 mice cortices from wild-type or NACHO knockout mice (TMEM35a$^{tm1(KOMP)Vlcg}$)[11] (obtained from the laboratory's vivarium). Cortices were dissociated using 10 U/mL papain (Worthington) digestion for 10 min, followed by trituration with a 10 mL glass pipette. Neurons were seeded at 15,000 cells per well (384-well plates) and maintained in NbActiv4® media (BrainBits®). Immunostaining or FLIPR assays were performed at DIV 20 or DIV 13, respectively, following 6 days incubation in nicotine (100 μM) or DFMO (5 mM) as indicated. For radioligand binding assays, neurons were transduced with SAT1 or α4 and β2 encoding lentiviruses (DIV 7) or treated with DFMO (DIV 14) and harvested on DIV 20. The lentiviral vectors encoding SAT1, α4 and β2 used a PGK promoter for transgene expression. The SAT1-expressing viral particles were packaged by Vigene Biosciences™, while the α4-expressing and β2-expressing viruses were packaged by VectorBuilder.

**Broad cDNA library screen**. The cDNA library used for α4β2 screening was from the Broad Institute and contains 5943 cDNA clones. This genome-wide screen used a FLIPR assay on transfected HEK293T cells in 384-well plates. Each well was transfected with 60 ng of total cDNA with the following composition: α4:β2:single Broad gene (2:4:3). Transfections with NACHO (α4:β2:NACHO [2:4:3]) served as positive control.

**FLIPR assay for agonist-evoked Ca$^{2+}$ influx**. High-throughput FLIPR assays were conducted using 384-well BioCoat (Corning) plates. Cells were washed briefly in assay buffer (HyClone™ HEPES-buffered saline [GE Life Sciences] comprising 149 mM NaCl, 4 mM KCl, 10 mM HEPES, and 5 mM glucose at pH 7.4 and 300 mOsm osmolality, supplemented with 2 mM CaCl$_2$ and 1 mM MgCl$_2$) prior to Calcium5 dye (Molecular Devices) loading for 1 h at RT. Following removal of excess dye, plates were placed in the FLIPR Tetra (Molecular Devices) chamber, and fluorescent Ca$^{2+}$ signal was captured using ScreenWorks 4.0™ software (Molecular Devices).

To obtain neuronal α7 mediated FLIPR responses, we used 5 μM PNU-12059615—a selective drug that attenuates receptor desensitization. Tetrodotoxin, or TTX (500 nM) was included in the assay buffer to inhibit spontaneous action potentials.

**Immunofluorescent staining and image analysis**. HEK293T cells seeded on 384-well BioCoat plates (Corning) were incubated with primary antibody against HA tag—Dylight 650® (Invitrogen™) in culture media for 1 h at 37 °C. Cells were fixed with 4% PFA (in HyClone™ HEPES-buffered saline) for 60 min, and after washing, nuclei were stained with DAPI or NucBlue® reagent (Invitrogen™).

For neurons, cells were fixed with 4% PFA for 1 h and then stained with fluorescent α-Bgt conjugates (1 μg/mL, AlexaFluor 647, Invitrogen™, B35450) or primary antibodies for 1 h. Then cells were simultaneous permeabilized and blocked in 0.2% Triton X-100 and 10% normal goat serum for 30 min. To label intracellular components, neurons were incubated in primary and secondary antibodies sequentially for 1 h each. Nuclei were stained with DAPI or NucBlue® reagent (Invitrogen™) prior to imaging.

Primary antibodies used were: HA (Mouse, 1:500, Invitrogen™ 2-2.2.14) conjugated to DyLight 650, GluA1 N-terminal (Mouse, 1:250, Millipore-Sigma RH95), Cytochrome C (Mouse, 1:100, Invitrogen™ 7H8.2C12), MAP2 (Chicken, 1:1000, Millipore-Sigma AB15452). Secondary antibody used: goat anti-mouse IgG (H + L) AlexaFluor 488 (Invitrogen™, A-11001), goat anti-chicken IgY (H + L) AlexaFluor 555 (Invitrogen™, A-21437) at 1:1000 dilution.

For quantification, images were acquired using Harmony™ high-content imaging software on an Opera Phenix™ screening instrument (PerkinElmer) with a ×20 objective. Data were further analyzed with the Columbus™ data storage and analysis system (PerkinElmer). Viable cells were identified from the DAPI signal (area >30 μm$^2$ for HEK293T; area >10 μm$^2$ for neurons). Cell perimeter was determined from the nuclear staining, and area of the cytoplasm was calculated based on the cytoplasmic region of interest. Average fluorescence intensity of wells was determined after subtracting the background signal, which was defined as anti-HA labeling of untransfected HEK293T cells or as α-Bgt647 labeling not displaced by 10 μM epibatidine.

**Radioligand binding assays**. HEK293T cells or cortical neurons were harvested in 50 mM ice-cold TrisHCl buffer (pH 7.4). Cells were homogenized for 30 s using the T-25 Ultra-Turrax homogenizer (Ika) and total protein concentration of the homogenate was determined using the Pierce™ BCA Protein Assay Kit (Thermo Scientific). Cell homogenates were incubated with 10 nM [³H]epibatidine (for nAChR) or with 30 nM [³H]flunitrazepam (for GABA) in 96-well plates for 3 h at room temperature. Nonspecific binding was determined by co-incubation of the cell samples with 10 μM unlabeled epibatidine or 100 μM unlabeled flunitrazepam. Assays were terminated by filtration through polyethylenimine-treated 96-well Unifilter GF/B plates (PerkinElmer). Filter plates were washed with 500 mL TrisHCL buffer and then desiccated at 65 °C for 30 min. MicroScint-0 scintillant

cocktail (50 μL; PerkinElmer) was added to each well and plates read with a TopCount NXT scintillation counter (PerkinElmer).

**Electrophysiology**. HEK293T cells were seeded (1 million/well) on uncoated six-well plates and transfected with cDNA combinations (total 2 μg/well) using FuGENE®6 transfection reagent (Promega Corporation). eGFP plasmid (10% of total cDNA) identified transfected cells. After 24 h, cells were dissociated using CellStripper™ dissociation reagent (Corning) and re-seeded on 12 mm glass coverslips (40,000/well). Electrophysiological recordings were done 48 h after transfection using external solution composed of HyClone™ HEPES-buffered saline (149 mM NaCl, 4 mM KCl, 10 mM HEPES, 2 mM CaCl$_2$, 1 mM MgCl$_2$ and 5 mM Glucose at pH 7.4; 300 mOsm osmolality). Intracellular solution contained (mM): 140 potassium gluconate, 10 HEPES, 4 Mg-ATP, 0.4 Na-GTP, and 0.6 EGTA (pH 7.3). To study most receptors, fast perfusion of compounds was achieved with the Perfusion Fast-Step system (Warner Instruments). To study α7 nAChRs, ultra-fast perfusion of compounds was achieved with a piezo-driven perfusion system and theta glass (Siskiyou) on to eGFP-expressing cells. The membrane holding potential was −70 mV. All recordings were performed at room temperature using an Axopatch 200B amplifier (Axon Instruments) and signals were filtered at 2 kHz and digitized at 10 kHz. For α7 nAChRs, signals were filtered at 10 kHz with a digitization rate of 50 kHz. Data acquisition and subsequent analysis were done with pClamp9 software (Axon Instruments).

**Statistics**. Results are represented as mean ± SD unless stated otherwise. All FLIPR assay, immunostaining experiments and radioligand binding assay in HEK293T cells and rat neurons were replicated thrice. Significance analyses between two datasets were performed with nonparametric Mann–Whitney $U$ test, while statistical analyses between three or more datasets used one-way ANOVA (GraphPad Prism, Carlsbad, CA). Significance level α = 0.05 was set.

**Reporting summary**. Further information on research design is available in the Nature Research Reporting Summary linked to this article.

## Data availability
Data supporting the findings of this manuscript are available from the corresponding author upon reasonable request. A reporting summary for this Article is available as a Supplementary Information file. The source data underlying Figs. 1b, d; 2b–d; 3b, c, g, i; 5b, c, e; 6b–e; 7c–e and Supplementary Figs. 2b; 3b, c; 4b, c, f; 5d, e; 6b–d are provided in the Source Data File. Source data are provided with this paper.

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

## Acknowledgements

The authors are grateful to Wes Davini for his valuable inputs.

## Author contributions

M.D. and M.L. performed cDNA library screening. J.A.M. conducted electrophysiology experiments. M.D. and H.Y. performed immunocytochemistry assays. M.D. conducted radioligand binding assays. D.K. and S.G. provided chimeric cDNA constructs. M.D. and D.S.B. wrote the manuscript. All authors contributed to the discussion and editing of the paper. D.S.B. supervised the project.

## Competing interests

All contributing authors are full-time employees in Johnson and Johnson.
