## [Peer Review File · Nature Communications]

Reviewers' Comments:

Reviewer #1:

Remarks to the Author:

Novel genome-wide screening was used to discover an unlikely candidate for selectively and efficiently promoting expression of alpha 7 and alpha 4 beta 2 AChRs through a novel mechanism. The small cytoplasmic protein SAT1 encodes the rate limiting enzyme for polyamine catabolism, and degrading polyamines turns out to promote assembly of these AChRs through interactions with their large cytoplasmic domains. The polyamine spermine can block the AChR cation channel, but does not alter the up regulation of AChRs by SAT1. Upregulation by SAT 1 depends on the presence of the alpha 4 or alpha 7 AChR large cytoplasmic domain. A cell-permeant inhibitor of SAT1 can block its effects. Blocking polyamine synthesis with DMFO (a drug that inhibits polyamide synthesis) was shown to up regulate alpha 4 beta 2 and alpha 7 AChR amounts and thereby nicotine-mediated protection of postsynaptic neurons from excitotoxicity, thereby suggesting a new approach to developing neuroprotective drugs. Mutations that reduce SAT1 expression are associated with increased suicide. Alpha 7 and alpha 4 beta 2 AChR agonists are thought to improve depressive behavior in patients and animal models.

These investigations have significant implications for drug development. These investigations also reveal previously unknown mechanisms for regulating expression of these AChR subtypes. The data presented are extensive, clearly presented, and compelling.

Reviewer #2:

Remarks to the Author:

Dhara et al show for the first time a new function of polyamines, distinct from that previously known of controlling channel gating.

They demonstrate that the intracellular reduction of polyamine levels, by means of different approaches, increases the assembly and plasma membrane trafficking of the alpha7 and alpha 4beta2 subtypes. They also clearly demonstrate that the polyamine effects on assembly involve the M3-M4 subunit cytosolic loop.

The effect of polyamine on nAChRs is present in neurons and the polyamine-induced nAChR upregulation is protective towards glutamate induce excitotoxicity; however, this protective effect is not present in neurons of NACHO KO mice.

The work is well done, clearly described and discussed.

The results are very interesting and deserve publication, although the cellular mechanisms acting on the different subtypes are not completely elucidated

Minor but important points

1) the title does not reflect properly the content of the paper. In the title there is no mention of nicotinic receptor subtypes, but in this paper the effect of polyamines is only on some of these subtypes. Indeed the reference to neuropharmacology is too vague.

2) In the introduction it is written that "polyamine levels selectively up-regulates cerebrocortical $\alpha 4\beta 2$ and $\alpha 7$ levels". Whereas in neurons it has been shown, by using non permeable α Bgtx647, an increase in alpha7-containing receptors, in the case of alpha4beta2 receptors this has not been demonstrated. The increase in 3H-epibatidine binding seen in figure 6d may be due to binding to both receptor subtypes. It will be important to perform the binding of 3H-Epipatidine in presence and absence of 100 nM Cytisine (that inhibits the binding to the alpha4beta2 subtype), in order to quantify in neurons the

increase in alpha7 and alpha4beta2 subtypes, respectively. Moreover by performing in neurons the Ca⁺ response in presence of an alpha4beta2 specific allosteric modulator it is also possible to know whether the functional alpha4beta2 receptors are increased at the cell surface.

3) At page 9 in discussion it is written that "Nicotine and other orthosteric ligands promote assembly through $\alpha 7$'s extracellular ACh binding domain⁴³". Please delete "extracellular" because Lester et al (43) show that the assembly is favoured by the intracellular action of nicotine and orthosteric ligands.

Point-by-point response to the referees

We have addressed the reviewers' concerns as documented below.

Reviewer #1. The reviewer wrote the following:

“Novel genome-wide screening was used to discover an unlikely candidate for selectively and efficiently promoting expression of alpha 7 and alpha 4 beta 2 AChRs through a novel mechanism. The small cytoplasmic protein SAT1 encodes the rate limiting enzyme for polyamine catabolism, and degrading polyamines turns out to promote assembly of these AChRs through interactions with their large cytoplasmic domains. The polyamine spermine can block the AChR cation channel, but does not alter the up regulation of AChRs by SAT1. Upregulation by SAT 1 depends on the presence of the alpha 4 or alpha 7 AChR large cytoplasmic domain. A cell-permeant inhibitor of SAT1 can block its effects. Blocking polyamine synthesis with DMFO (a drug that inhibits polyamide synthesis) was shown to up regulate alpha 4 beta 2 and alpha 7 AChR amounts and thereby nicotine-mediated protection of postsynaptic neurons from excitotoxicity, thereby suggesting a new approach to developing neuroprotective drugs. Mutations that reduce SAT1 expression are associated with increased suicide. Alpha 7 and alpha 4 beta 2 AChR agonists are thought to improve depressive behavior in patients and animal models.

These investigations have significant implications for drug development. These investigations also reveal previously unknown mechanisms for regulating expression of these AChR subtypes. The data presented are extensive, clearly presented, and compelling.”

Authors' response: This reviewer had no concerns with our manuscript. We appreciate this reviewer's laudatory comments.

Reviewer #2. The reviewer wrote the following:

“Dhara et al show for the first time a new function of polyamines, distinct from that previously known of controlling channel gating.

They demonstrate that the intracellular reduction of polyamine levels, by means of different approaches, increases the assembly and plasma membrane trafficking of the alpha7 and alpha 4beta2 subtypes. They also clearly demonstrate that the polyamine effects on assembly involve the M3-M4 subunit cytosolic loop.

The effect of polyamine on nAChRs is present in neurons and the polyamine-induced nAChR upregulation is protective towards glutamate induce excitotoxicity; however, this protective effect is not present in neurons of NACHO KO mice.

The work is well done, clearly described and discussed."

The reviewer also noted that *"The results are very interesting and deserve publication, although the cellular mechanisms acting on the different subtypes are not completely elucidated."*

Authors' response: To address this issue, we considered that the TM3-TM4 loop determines polyamine-mediated assembly. Accordingly, we aligned these sequences for $\alpha 7$, $\alpha 4$ and $\alpha 6$ nAChR subunits. Although the sequences show limited similarities between subunits, certain negatively charged residues in the TM4 proximal portion of the TM3-TM4 loop are conserved between $\alpha 7$ and $\alpha 4$ but are divergent in $\alpha 6$. Strikingly, mutating those negatively charged residues to non-polar amino acid in $\alpha 7$ or $\alpha 4$ leads to loss of SAT1 mediated upregulation of receptor expression. Similarly, introducing negative charges at similar position of the $\alpha 6$ cytosolic loop leads to significant SAT1 and DFMO dependent increase in $\alpha 6\beta 4$ surface expression. Taken together, these new experiments identify crucial residues within the TM3-TM4 cytosolic loop that can determine polyamine regulation of specific nAChR. These data are presented in Supplementary Fig. 5.

The reviewer listed three minor points that we addressed as follows:

1. The reviewer commented that *"1) the title does not reflect properly the content of the paper. In the title there is no mention of nicotinic receptor subtypes, but in this paper the effect of polyamines is only on some of these subtypes. Indeed the reference to neuropharmacology is too vague."*

Authors' response: This is a fair comment, we propose changing the title to "Polyamine regulation of ion channel assembly: implications for nicotinic acetylcholine receptor pharmacology".

2. The reviewer wrote:

2) In the introduction it is written that "polyamine levels selectively up-regulates cerebrocortical $\alpha 4\beta 2$ and $\alpha 7$ levels". Whereas in neurons it has been shown, by using non permeable $\alpha Bgtx647$, an increase in alpha7-containing receptors, in the case of alpha4beta2 receptors this has not been demonstrated. The increase in 3H-epibatidine binding seen in figure 6d may be due to binding to both receptor subtypes. It will be important to perform the binding of 3H-Epibatidine in presence and absence of 100 nM Cytisine (that inhibits the binding to the alpha4beta2 subtype), in order to quantify in neurons the increase in alpha7 and alpha4beta2 subtypes, respectively.

Authors' response: This is an important consideration. Previous studies have shown that genetic ablation of $\alpha 4$ results in loss of [^3H]epibatidine binding throughout the brain, including cortical areas (Marubio et al., 1999). In contrast, [^3H]epibatidine binding is not significantly altered in brain tissues in $\alpha 7$ knockout mice (Franceschini et al., 2002). Thus, our [^3H]epibatidine binding assay in cortical neurons (14.97 ± 2.0 fmol/mg) (Fig. 6d) almost exclusively reflects binding to $\alpha 4\beta 2$ receptors and this is enhanced with DFMO pretreatment (34.76 ± 6.1 fmol/mg) or viral transduction of SAT1 (32.62 ± 7.4 fmol/mg). To further support our conclusion that polyamines can control assembly of neuronal $\alpha 4\beta 2$ receptors, we transduced cortical neurons with $\alpha 4$ and $\beta 2$ encoding lentiviruses. [^3H]Epibatidine binding to the transduced neuronal membranes (290.39 ± 30.4 fmol/mg) was almost 20 times higher than untransduced neurons. Importantly, DFMO pretreatment showed a similar magnitude increase in binding (~ two-fold, 521.94 ± 47 fmol/mg). Thus, polyamine levels can clearly regulate neuronal $\alpha 4\beta 2$. These new neuronal experiments are included in Fig. 6d.

The reviewer also noted that *“Moreover, by performing in neurons the Ca^{2+} response in presence of an alpha4beta2 specific allosteric modulator, it is also possible to know whether the functional alpha4beta2 receptors are increased at the cell surface.”*

Authors' response: In fact, we have tested several $\alpha 4\beta 2$ positive allosteric modulators (PAMs). As previously published (dFBr, Sala et al., 2005; LY 2087101, Broad et al., 2006), we find that these PAMs nicely potentiate $\alpha 4\beta 2$ in heterologous cells and in *Xenopus* oocytes. Unfortunately, we find that these compounds do not potentiate nAChR activity as measured by FLIPR in neurons. Indeed, we could not find published success with these compounds in neuronal FLIPR experiments.

3. The reviewer wrote, *“(3) At page 9 in discussion it is written that “Nicotine and other orthosteric ligands promote assembly through $\alpha 7$'s extracellular ACh binding domain43”. Please delete “extracellular” because Lester et al (43) show that the assembly is favoured by the intracellular action of nicotine and orthosteric ligands.”*

Authors' response: As suggested, on page 9 of the discussion we deleted “extracellular” when referring to the site the action of orthosteric ligands to promote $\alpha 7$ assembly.

We thank the reviewers for their time and insightful recommendations, which have served to substantially improve this revised manuscript. We hope it is now suitable for publication in *Nature Communications*.

REFERENCES

1. Marubio LM., del Mar Arroyo-Jimenez M., Cordero-Erausquin M., Léna C., Le Novère N., de Kerchove d'Exaerde A., Huchet M., Damaj MI., & Changeux JP. Reduced antinociception in mice lacking neuronal nicotinic receptor subunits. *Nature* **398**, 805-810 (1999).
2. Franceschini D., Paylor R., Broide R., Salas R., Bassetto L., Gotti C., & De Biasi M. Absence of alpha7-containing neuronal nicotinic acetylcholine receptors does not prevent nicotine-induced seizures. *Brain Res Mol Brain Res.* **98**, 29-40. (2002).
3. Sala F., Mulet J., Reddy KP., Bernal JA., Wikman P., Valor LM., Peters L., König GM., Criado M., & Sala S. Potentiation of human $\alpha 4\beta 2$ neuronal nicotinic receptors by a *Flustra foliacea* metabolite. *Neurosci.Lett.* **373**, 44 (2005).
4. Broad LM., Zwart R., Pearson KH., Lee M., Wallace L., McPhie GI., Emkey R., Hollinshead SP., Dell CP., Baker SR., & Sher E. Identification and pharmacological profile of a new class of selective nicotinic acetylcholine receptor potentiators. *J Pharmacol Exp Ther.* **318**, 1108-1117 (2006).

Reviewers' Comments:

Reviewer #1:

Remarks to the Author:

This paper reports novel discoveries about the role of polyamines in regulating the expression of alpha 7 and alpha 4 beta 2 AChRs but not alpha 6 beta 4 AChRs. Extensive data are provided. The authors were responsive to reviewer comments. This is an interesting and important paper with substantial implications for new therapeutic approaches.

Reviewer #2:

Remarks to the Author:

The authors have addressed all the issues that I have raised and inserted them in the revised version of the paper.